# Viral Ecogenomics of Arctic Cryopeg Brine and Sea Ice

Zhi-Ping Zhong,[a,b] Josephine Z. Rapp,[c] James M. Wainaina,[b] Natalie E. Solonenko,[b] Heather Maughan,[d] Shelly D. Carpenter,[c] Zachary S. Cooper,[c] Ho Bin Jang,[b] Benjamin Bolduc,[b] Jody W. Deming,[c] Matthew B. Sullivan[a,b,e,f]

aByrd Polar and Climate Research Center, The Ohio State University, Columbus, Ohio, USA
bDepartment of Microbiology, The Ohio State University, Columbus, Ohio, USA
cSchool of Oceanography, University of Washington, Seattle, Washington, USA
dRonin Institute, Montclair, New Jersey, USA
eDepartment of Civil, Environmental and Geodetic Engineering, The Ohio State University, Columbus, Ohio, USA
fCenter of Microbiome Science, The Ohio State University, Columbus, Ohio, USA

**ABSTRACT** Arctic regions, which are changing rapidly as they warm 2 to 3 times faster than the global average, still retain microbial habitats that serve as natural laboratories for understanding mechanisms of microbial adaptation to extreme conditions. Seawater-derived brines within both sea ice (sea-ice brine) and ancient layers of permafrost (cryopeg brine) support diverse microbes adapted to subzero temperatures and high salinities, yet little is known about viruses in these extreme environments, which, if analogous to other systems, could play important evolutionary and ecosystem roles. Here, we characterized viral communities and their functions in samples of cryopeg brine, sea-ice brine, and melted sea ice. Viral abundance was high in cryopeg brine ($1.2 \times 10^8$ ml$^{-1}$) and much lower in sea-ice brine ($1.3 \times 10^5$ to $2.1 \times 10^5$ ml$^{-1}$), which roughly paralleled the differences in cell concentrations in these samples. Five low-input, quantitative viral metagenomes were sequenced to yield 476 viral populations (i.e., species level; $\geq$10 kb), only 12% of which could be assigned taxonomy by traditional database approaches, indicating a high degree of novelty. Additional analyses revealed that these viruses: (i) formed communities that differed between sample type and vertically with sea-ice depth; (ii) infected hosts that dominated these extreme ecosystems, including *Marinobacter*, *Glaciecola*, and *Colwellia*; and (iii) encoded fatty acid desaturase (*FAD*) genes that likely helped their hosts overcome cold and salt stress during infection, as well as mediated horizontal gene transfer of *FAD* genes between microbes. Together, these findings contribute to understanding viral abundances and communities and how viruses impact their microbial hosts in subzero brines and sea ice.

**IMPORTANCE** This study explores viral community structure and function in remote and extreme Arctic environments, including subzero brines within marine layers of permafrost and sea ice, using a modern viral ecogenomics toolkit for the first time. In addition to providing foundational data sets for these climate-threatened habitats, we found evidence that the viruses had habitat specificity, infected dominant microbial hosts, encoded host-derived metabolic genes, and mediated horizontal gene transfer among hosts. These results advance our understanding of the virosphere and how viruses influence extreme ecosystems. More broadly, the evidence that virally mediated gene transfers may be limited by host range in these extreme habitats contributes to a mechanistic understanding of genetic exchange among microbes under stressful conditions in other systems.

**KEYWORDS** viral communities, extreme environments, virus-host interaction, cold and salt adaption, horizontal gene transfer

Address correspondence to Jody W. Deming, jdeming@uw.edu, or Matthew B. Sullivan, sullivan.948@osu.edu.

mSystems®

The cryosphere includes those regions of the Earth's surface with frozen water, including snow cover, glaciers, ice sheets and shelves, icebergs, freshwater ice, sea ice, permafrost, and ground ice (1). Predominantly found in polar regions, these frozen seascapes and landscapes are strongly affected by global warming (2, 3), with temperatures in the Arctic rising two to three times faster than the global average (4). This warming has resulted in an unprecedented reduction of sea ice, with mean September sea-ice extent declining ~14% per decade since 1979 (5) and permafrost in northern Sweden, for example, being lost at a rate of ~1 cm per year since 1978 (6). Such ecosystem losses are also shrinking our opportunity to understand the resident organisms and their unique adaptations to these extreme habitats and how they will respond to climate change.

A major challenge in assessing microbial responses of such high-latitude ecosystems is the highly varied niche space that exists within them under extreme conditions. Each of these frozen environments contains inhabitable brines that remain liquid below the freezing point of water due to their high salt concentrations. Despite extreme conditions, both sea ice (e.g., see references 7 and 8) and cryopegs (relic marine sediments bounded by permafrost; e.g., see references 9–11) contain brines with abundant and diverse microbial communities. Relative to that of seawater, bulk sea ice is reduced in microbial cell density, while the liquid brines within sea ice (sea-ice brine) are concentrated with respect to microbes, salts, and other solutes and particles (organic and inorganic) present in the source water at the time of freezing (8, 12). Cryopegs, which are ancient marine sedimentary systems that contain lenses of hypersaline brine within year-round Arctic frozen soils (also called permafrost) (11, 13), are also habitable by having a volume of brine that remains liquid at subzero temperature due to high salt concentrations (13–15). Both sea-ice and cryopeg brines can be similar in their extreme temperature and salinity conditions and in microbial density, which can enhance the contact rates and, thus, gene exchange between viruses and bacteria (15, 16). However, they differ substantially in formation process, stability, and age: sea-ice brines derive from surface seawater, fluctuating in their properties during daily-to-annual freeze-thaw cycles, while cryopeg brines are thought to derive from relic seawater trapped in marine (15) or estuarine (17, 18) sediments from the late Pleistocene, where they have been stable and isolated from the atmosphere for many thousands of years.

Microbial communities in sea ice and cryopegs have been investigated previously in many studies. Taxonomically, *Proteobacteria* and *Bacteroidetes* dominate PCR-based amplicon sequencing of both types of brine (7, 9). For example, in one comparative study, cryopeg brines were dominated by *Gammaproteobacteria* and *Bacteroidia*, with average relative abundances of 57% and 22%, respectively, while sea-ice brines were dominated by *Bacteroidia* (average relative abundance of 51%) followed by *Gammaproteobacteria* (average relative abundance of 30%) (9). Although some microbes may only be dormant in these systems (19), studies have documented active microbial communities in sea ice (reviewed in reference 8) and cryopeg brines (9, 11, 15). Thus, at least some microbes in these environments have adapted to the extremely cold and saline conditions and are contributing to ecosystem functioning and biogeochemical cycling.

Viruses in brines, however, are less well studied, although in other environments they play key ecosystem roles. For example, in low-latitude marine systems, marine viruses are abundant ($10^6$ to $10^9$ particles ml$^{-1}$ [20, 21]) and can alter microbial community structure, diversity, and function through lysis, horizontal gene transfer (HGT), and metabolic reprogramming (22–25). Similar findings are emerging from other systems, such as permafrost soils (26), deep-subsurface fracking wells (27), animal rumens (28, 29), human gut (30), and freshwater environments (31). Although viruses are known to be abundant in sea ice (16, 19, 32), cryopeg brines (15), and other subzero brines (as summarized in reference 15 and Table 2 therein), no study has used a modern viral ecogenomics toolkit (22) to study the viral communities and their ecological role in these extreme environments, due in part to limited sample volumes and availability. If these other ecosystems are any indication, then viruses that infect microbes in subzero brines are also likely to play key roles in these extreme ecosystems.

Here, we leveraged low-input, quantitative sample-to-sequence metagenomic approaches (33–38) to obtain viral ecogenomic data from the viral fraction of five cryosphere samples, including cryopeg and sea-ice brine and bulk sea ice collected near Utqiaġvik, Alaska. We used these data to assess viral abundances and communities and to determine whether viruses in such extreme subzero hypersaline habitats impact their microbial hosts through modulating host genes and mediating HGT of key genes among hosts.

## RESULTS AND DISCUSSION

**Viromes from melted sea ice, sea-ice brine, and cryopeg brine.** In May 2017, we collected five samples near Utqiaġvik, Alaska: one cryopeg brine sample from below the floor of the Barrow Permafrost Tunnel (9); three samples from sections of a sea-ice core (upper sea ice, middle sea ice, and lower sea ice) in landfast first-year sea ice; and one sample of sea-ice brine drained into a sackhole in the sea ice (Fig. 1). Counts of virus-like particles (VLP) indicated that the cryopeg brine had $1.2 \times 10^8$ VLP $ml^{-1}$, while VLP counts from sea-ice brine and bulk (melted) sea ice were much lower ($1.2 \times 10^5$ to $2.1 \times 10^5$ VLP $ml^{-1}$ [39]); thus, we could not address whether viruses were concentrated in the liquid phase (brine) of sea ice (see Table S1 in the supplemental material in reference 40). Similarly, the concentrations of viral DNA extracted from the sea ice-derived samples were also below the detection limit ($<0.1$ ng/$\mu$l) of Qubit 3.0 (Thermo Fisher Scientific). Therefore, all samples were subjected to low-input, quantitative viral metagenomic sequencing, which can recover viruses from samples with DNA concentrations as low as 100 fg/$\mu$l (35–37). The recovered DNA was used for short-read metagenomic sequencing library construction (see Materials and Methods) to yield five viral metagenomes (viromes) with a total of 355,904,145 reads (Table S2 [40]). The quality-controlled reads of each sample were assembled individually, resulting in 7,404 and 2,577 contigs of $\geq$5 and $\geq$10 kb, respectively, across the full data set (Table S2 [40]).

VirSorter (41) predicted a total of 1,692 viral contigs of $\geq$5 kb in size, almost all of which (1,686 of 1,692; categories 1, 2, and 3) were not identified as a prophage and so were interpreted to derive from extrachromosomal prophages, lytic infections, or partially assembled prophages (Table S2 [40]) (41). Of these 1,686 contigs, most (1,393) were high-confidence viruses belonging to category 1 or 2 that we then dereplicated into 1,305 viral populations using currently accepted cutoffs that approximate species-level taxonomy (42–45). On average, about 20.0% (range of 1.3 to 70.1% for five viromes) of the quality-controlled reads were recruited to these viral populations (Table S2 [40]), which were about 2-fold higher than the typical read recruitments in the Global Ocean Viromes data set (23, 42). From each of the 1,305 populations the longest contig was selected for all further investigations, except for the taxonomic and host predictions, which were restricted to the 476 populations with sequences of $\geq$10 kb in length. Rarefaction curves were constructed (see Materials and Methods) and showed that the sequencing depth of the cryopeg sample was sufficient to capture the diversity of long viral populations present, and that the sea-ice viromes were slightly undersampled (Fig. S1 [40]). Thus, additional, unsampled viral populations likely exist in the sea-ice environment.

**Viral communities comprise mostly novel genera and differ across habitats.** We next considered how viruses in these underexplored extreme environments compared to known viruses. Because viruses lack any single, universally shared gene, we established taxonomy using gene-sharing analysis from viral sequences $\geq$10 kb in length using vConTACT v2 (46, 47). In our data set, that meant comparing shared gene sets from 476 viral populations (166 from cryopeg plus 310 from sea-ice populations) to genomes from 2,304 known viruses in the NCBI RefSeq database (version 85) (Fig. 2). Such analyses produce viral clusters (VCs), which represent approximately genus-level taxonomic assignments (46, 47). The 2,304 RefSeq viral genomes formed 370 VCs (Table S3 [40]). Of the 476 viral populations in our data, 255 (83 cryopeg plus 172 sea ice) were grouped into 97 genera/VCs (VC with $\geq$2 viruses), while only 56 (33 cryopeg plus 23 sea

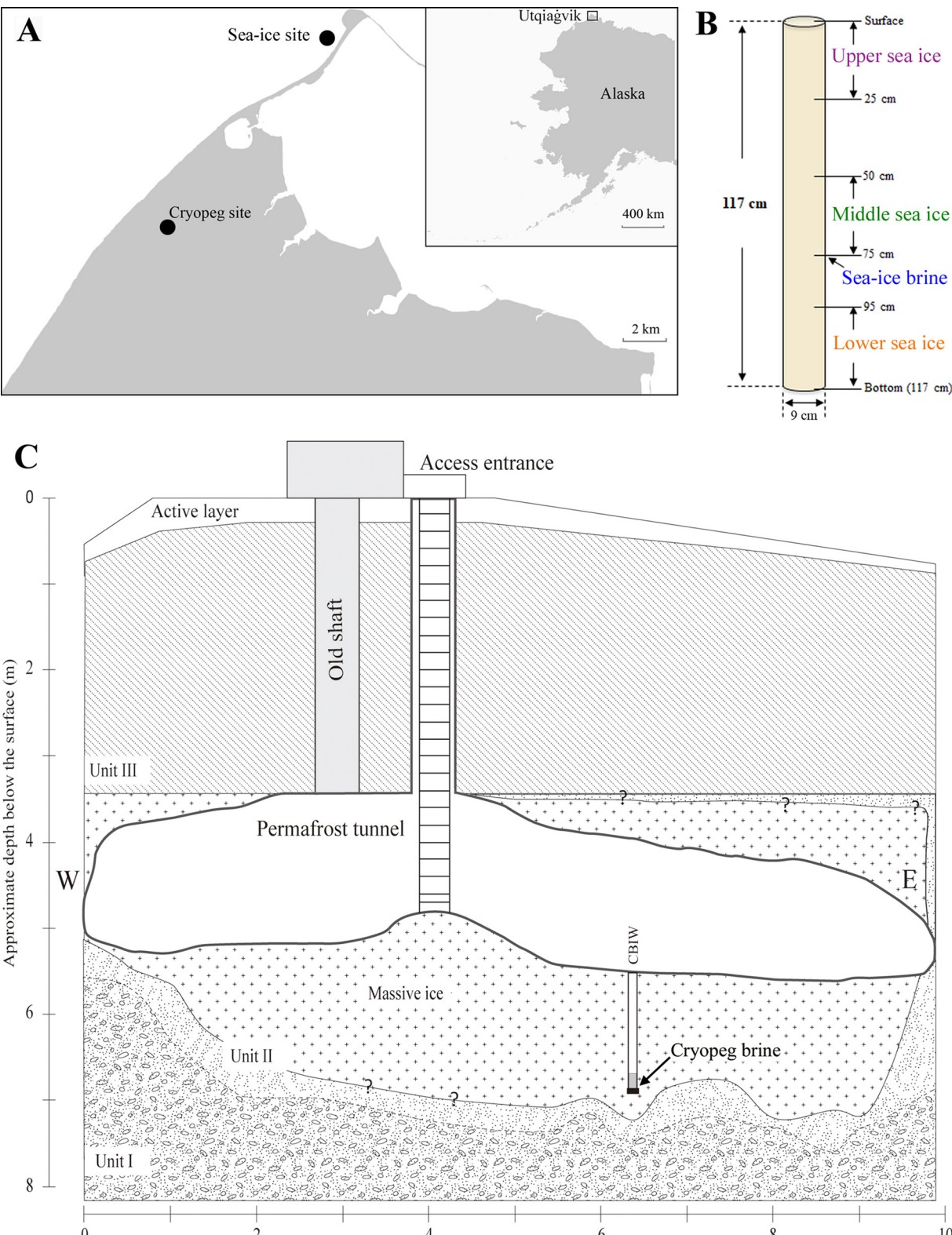

**FIG 1** Sampling of sea ice and cryopeg samples near Utqiaġvik, Alaska. (A) Location of sampling sites near Utqiaġvik, Alaska (inset). (B) Schematic of sampling depths of the four samples obtained from sea ice. The arrows indicate the ice sections from an ice core collected for upper, middle, and lower sea ice, as well as the depth (i.e., 75 cm) of the (separate) sackhole drilled for draining brine (sea-ice brine) from the sea ice. (C) Schematic of the Barrow Permafrost Tunnel and borehole access to the cryopeg brine reservoir. The cryopeg brine sample was collected about 1.5 m below the tunnel floor or 7 m below the surface. (Adapted from reference 9 with permission of the publisher.)

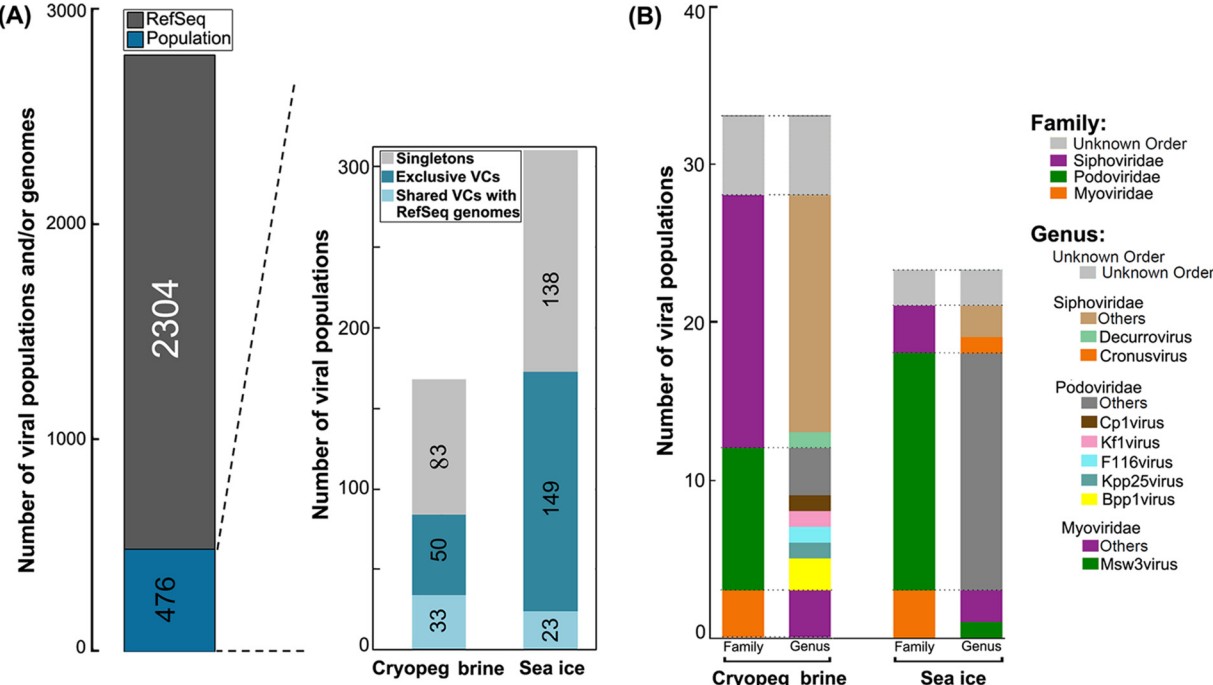

**FIG 2** Taxonomic assignments of viruses extracted from the field samples. (A) The bar graph on the left shows the numbers of viral populations and/or genomes used for taxonomic assignments from the field samples and RefSeq database. The bar graph on the right shows the numbers of viral populations detected in cryopeg brine and in the sea-ice samples. Data from the bulk sea-ice sections and sea-ice brine (Fig. S3 [40]) were combined to enable comparison between the two environments. Viral populations were classified into three groups of VCs: singletons (gray) that had no close relatives, exclusive VCs (aqua) that comprised field populations only, and shared VCs with RefSeq genomes (sky blue) that comprised field populations with taxonomy assignments and genomes from RefSeq. (B) Bar graph indicating the numbers and taxonomy (both family and genus levels) of viral populations from cryopeg and sea-ice environments. Viruses from the category "shared VCs with RefSeq genomes" are shown.

ice) of these 255 populations formed VCs (32 genera/VCs) with RefSeq viral genomes and could be assigned taxonomy. The other 221 (of 476) populations (83 cryopeg plus 138 sea ice) remained isolated as singletons (i.e., as single-member VC) due to having no or few shared genes with other viruses (Fig. 2). Together, only 12% (56 of 476) of viral populations could be assigned taxonomy. These results indicate that myriad unique viruses inhabit cryopeg brine and sea-ice environments. Comparing viral clusters in the cryopeg to those in the sea-ice samples showed that only five VCs (of 97 total VCs) were shared between the two sample types (Fig. S2 [40]), which implies largely distinct viral communities between cryopeg and sea-ice habitats. The shared VCs suggest similarities in viral reproduction, as genus-level viral communities typically reflect the broad characteristics of viral replication and packaging (46), as opposed to ecological distinctions (see below). Four of these five shared VCs contained viral populations exclusive to cryopeg brine and sea-ice environments (i.e., not available in the RefSeq database; Fig. S2 [40]), while the other one included sequences most similar to a temperate *Myxococcus* phage Mx8 (VC_248_0; Table S3 [40]) in the RefSeq database (48). We then determined the number of viral genera (i.e., VCs) in each sample by mapping QC reads to viral populations in each VC (see Materials and Methods). Cryopeg brine harbored 167 populations (representing 47 VCs), whereas the sea-ice samples contained 54 (30 VCs), 51 (24 VCs), 169 (48 VCs), and 255 (44 VCs) populations for the upper, middle, and lower sea ice and sea-ice brine, respectively (Fig. S3 [40]).

We then looked more closely at the population (approximately species) level to compare viral communities in the cryopeg and sea-ice environments. We first calculated the abundance of each of the 1,305 viral populations (≥5 kb in length) by read mapping (see Materials and Methods). The 100 most abundant populations (summed across all five samples) were used to compare viral community compositions in the five viromes by a heatmap (Fig. 3A). The results showed that the viral community in cryopeg

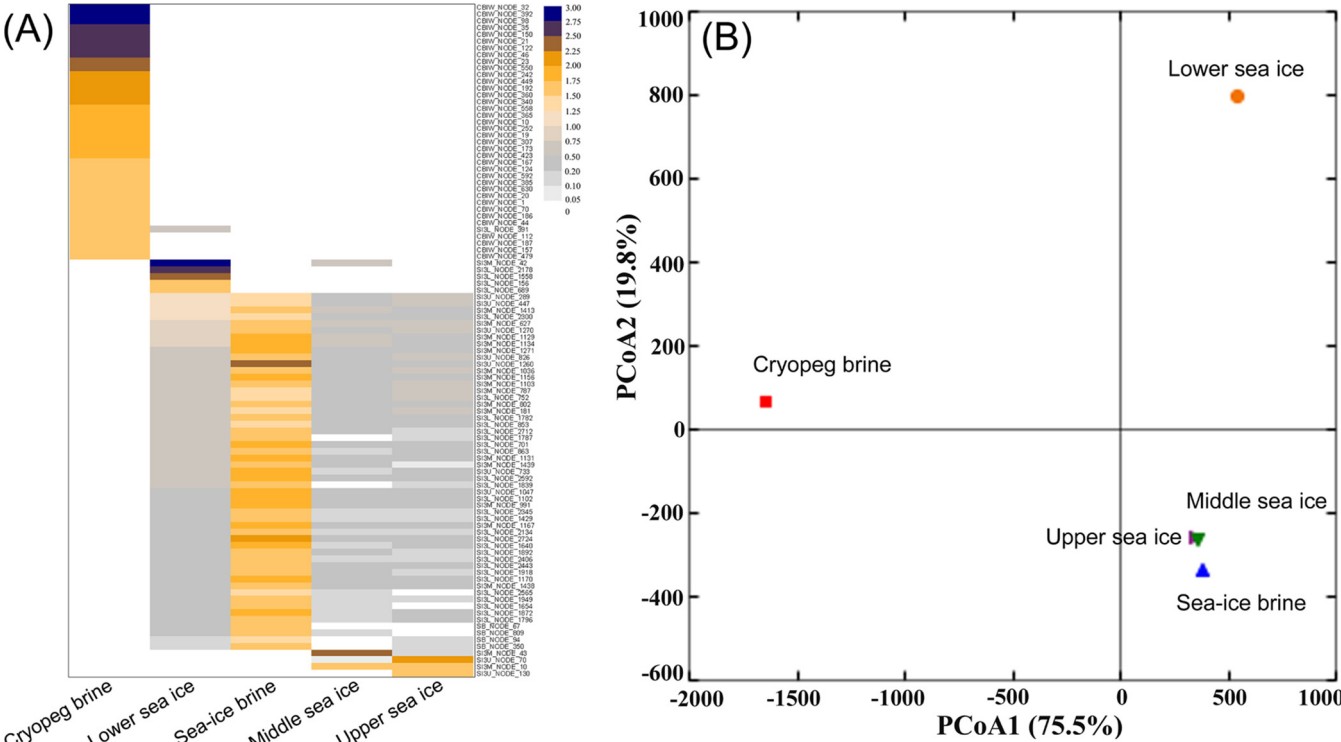

**FIG 3** Heatmap and principal coordinate analysis (PCoA) of viral community composition. (A) Heatmap showing the coverage of top 100 abundant viral populations (≥5 kb) per gigabase of MetaG (log$_{10}$) for five viromes. (B) PCoA plot of the cryopeg brine and sea-ice samples. For both PCoA plots, each symbol represents a sample as indicated on the plots. Viral community composition was generated based on the coverage of viral populations ($n$ = 1,305; length, ≥5 kb), which was determined by mapping quality-controlled reads to each viral population followed by normalization per gigabase of virome.

brine comprised different populations from those in the sea-ice brine and bulk sea-ice samples, with the exception of one shared population (Fig. 3A and Table S4 [40]), and that the four sea-ice samples shared most of their populations. Principal coordinate analysis (PCoA) of the viral population abundances also showed that the viral community in cryopeg brine was distinct from the sea-ice communities (Fig. 3B). From the sea-ice environment, the upper sea-ice, middle sea-ice, and sea-ice brine samples clustered together (Fig. 3B), whereas lower sea ice grouped separately from the sea-ice sample cluster.

These results indicate that the cryopeg brine supports a unique viral community (consistent with a previous study [15]), which may be due to this cryopeg system having been separated from the atmosphere and meteoric water for at least 14,000 years and possibly much longer (9, 17, 18). Viral communities inhabiting the middle and top layers of the ice core were similar to each other and to that of sea-ice brine, as expected given that this brine was drained from these two sea-ice layers. The separation can be attributed to the ice-algal bloom and associated bacteria, as well as some physiochemical factors in the bottom portion of the ice (i.e., high concentrations of chlorophyll, phaeopigment, bacteria, EPS, POC, and PON; Table S1 [40]). These results indicate that viral communities vary across vertical positions of the sea-ice core, which is consistent with previous reports on varied bacterial and microalgal communities at different ice depths (49, 50).

**Viruses infect dominant microbes in subzero and salty habitats.** We next examined microbial communities in these samples to identify potential hosts. Microbial community structures, generated by 16S rRNA gene amplicon sequencing (see Materials and Methods), showed all samples to be dominated by *Bacteroidia* of the phylum *Bacteroidetes* and *Gammaproteobacteria* of the phylum *Proteobacteria* (Fig. S4 and Table S5 [40]). Based on their most abundant genera, the microbial community in the cryopeg brine differed from those in the sea-ice environments (Fig. S4 [40]). While the cryopeg

community was dominated by *Marinobacter* (relative abundance of 53.0%), *Gillisia* (21.1%), and *Demequina* (7.7%), these groups were of minor relative abundance (0.1 to 1.1%) in the sea-ice samples (Fig. S4 [40]). Instead, sea-ice communities were dominated by *Paraglaciecola* (3.4 to 21.5%), *Reinekea* (1.2 to 12.8%), *Colwellia* (4.9 to 8.2%), *Polaribacter* (8.4 to 36.6%), and *Glaciecola* (0.5 to 2.5%), which were all rare ($\leq$0.1%) in the cryopeg brine (Fig. S4 [40]). Many of these abundant genera contain species that are halotolerant or psychrotolerant halophilic and/or psychrophilic, such as *Marinobacter halophilus* (51), *Colwellia psychrotropica* (52), and *Glaciecola punicea* (53). In addition, some members of *Marinobacter* can utilize aromatic and aliphatic hydrocarbons as sole carbon and energy sources (e.g., see references 54–56), and *Glaciecola* can produce cold-active and salt-tolerant enzymes to hydrolyze xylan and cellulose (e.g., see references 57–59). These results indicate that the bacterial communities in cryopeg brine and sea-ice ecosystems are dominated by taxa that are likely equipped with genomic adaptations to subzero and salt conditions and that play important carbon-cycling roles in these extreme environments.

We then explored the potential impacts of viruses on these abundant microbes by linking viruses *in silico* to their hosts. Hosts were predicted using two methods, one based on the viral and host nucleotide sequence similarity (60) and the other based on genome composition (61). Because the accuracy of host prediction increases with viral contig length (61), only the 476 viral populations of $\geq$10 kb in size were used for host prediction. The sequence similarity method (BLASTN) predicted hosts for only 13 populations (Table S6 [40]), whereas the sequence composition method (using the software VirHostMatcher [61]) linked many more populations ($n = 141$) to microbial hosts (Table S7 [40]). Hosts at higher taxonomic levels (i.e., phylum and class levels) were most reliably linked to populations, similar to previous reports (60, 61). Both methods consistently predicted phylum- and class-level hosts for the same ten populations and genus-level hosts for five of these ten populations (Table S6 and S7 [40]). Although only 29.6% of viral populations could be linked to a host, these host predictions indicated that viruses in cryopeg brine and sea-ice samples infect microbes inhabiting these extreme environments.

The predicted host phyla (identified by the sequence composition method) that were most abundant included *Proteobacteria*, *Bacteroidetes*, and *Actinobacteria* (Fig. S5 and Table S7 [40]). Interestingly, *Actinobacteria*-linked viruses were abundant (12.7%; Fig. S5 [40]) in cryopeg brine but were not detected in sea-ice environments (except at low abundance [0.3%] in sea-ice brine), indicating that *Actinobacteria*-associated viral predation happens more often in cryopeg brine than in sea-ice environments. This conclusion is consistent with the higher relative abundance of *Actinobacteria* in cryopeg brine (12.3%) than in sea-ice environments (0.4 to 2.5%; Fig. S5 [40]). The predicted host genera (identified by the sequence similarity method) that were most abundant in these samples included *Marinobacter*, *Glaciecola*, and *Colwellia* (Table S6 [40]), many members of which are tolerant of low temperature and high salinity, as discussed above. The relative abundance of *Marinobacter*-associated viral populations was high (12.9%) in cryopeg brine (Table S6 [40]), including one (i.e., CBIW_NODE_98) with the third highest coverage (coverage of 684) among all populations in this sample (Table S4 [40]). This finding is consistent with the dominance (53.0%) of the bacterial genus *Marinobacter* in this cryopeg brine sample (Fig. S4 [40]) and in other samples of cryopeg brine from the same location (9). The *Glaciecola*-associated virus SI3M_NODE_10 was the second most abundant (relative abundance, 14.8 to 18.0%) viral population in both middle and upper sea ice viromes (Tables S4 and S6 [40]), while members of *Glaciecola* only accounted for 1.4 to 1.9% of microbial profiles in these samples (Fig. S4 [40]). *Colwellia*-linked virus SI3M_NODE_1 was the sixth most abundant viral population and had a relative abundance of 1.3% in middle sea ice, where the microbial community was dominated by the genus *Colwellia*, with 8.3% relative abundance (Fig. S4 [40]). Given the high relative abundances of *Marinobacter*, *Glaciecola*, and *Colwellia*, as well as their associated viruses, we predict that these viruses influence the

responses of their hosts to cold and salty conditions and contribute to ecosystem function in these extreme environments.

Overall, the microbial profiles and host predictions suggested that viruses have infected the abundant microbial groups and, thus, might have an important role in cryopeg and sea-ice ecosystems by influencing their hosts.

**Virus-encoded cold and salt tolerance genes potentially impact their hosts.** We next sought to identify virus-encoded auxiliary metabolic genes (AMGs) that might directly modify host cold or salt adaption during infection. Briefly, 22,820 predicted genes from the 1,305 viral populations (length, ≥5 kb) were queried against functional databases, which resulted in about half (11,002) matching (bit score, ≥60) annotated sequences in KEGG, UniRef, or InterProScan databases (Table S8 [40]). Within these, 76 were identified as possible AMGs (see Materials and Methods), which derived from 293 viral genes in 138 viral populations (Table S9 [40]) and included 61 AMGs that were reported previously (23, 62). Among those not seen before, we explored whether they may offer new insights into how viruses manipulate microbial metabolisms. For example, 3 new AMGs included *Cellulase*, *CHB_HEX_C_1*, and *GDP_Man_Dehyd* that we hypothesized mediate sugar metabolisms in their hosts during infection (Table S9 [40]). However, we focused on fatty acid desaturase (*FAD*) genes due to their potential function in cellular tolerance of extreme conditions. Specifically, in diverse microbes, *FAD* genes are known to desaturate cell membrane lipids by introducing double bonds into the hydrocarbon chains of fatty acids (63), so as to increase membrane fluidity (64) to help cells cope with stresses, including low temperature (65, 66) and high salinity (67, 68).

From four viral populations, we identified 12 *FAD* genes (Table S9 [40]) that we had high confidence were encoded by viruses due to flanking viral genes (Fig. S6 [40]). We then determined the evolutionary history of 11 virus-encoded *FAD* genes (*vFAD*; one was removed due to potential recombination; see Materials and Methods) by phylogenetic analysis with 217 and 263 microbial *FAD* (*mFAD*) genes from the concurrently sampled microbial metagenomes and NCBI's nr database, respectively. This tree resolved *FAD* genes that belonged to the same bacterial or eukaryotic phylum into distinct clades (Fig. S7 [40]), which implied that these *FAD* genes had not been transferred across different bacterial phyla or between bacteria and eukaryotes. To look at this more closely, we used a subset of the *FAD* genes (see Materials and Methods) and found patterns similar to those observed in the above "all-sequence" tree, specifically, four clades of *vFAD*, which we designated A to D (Fig. 4). These results suggest that host-to-phage *FAD* transfers occurred at least four times, which is consistent with multiple transfers observed in cyanobacteria and their cyanophages for the AMGs *PstS* (69), *PsbA* and *PsbD* (70), and *PTOX* (71).

Interestingly, one sea-ice-originated *vFAD* gene (SB_NODE_339_gene_12) clustered with 17 *mFAD* genes from the *Proteobacteria* phylum in clade D (Fig. 4), which we interpret to represent a case of phage-mediated *FAD* gene transfers among *Proteobacteria* members. In addition, this *vFAD* sequence shared a high amino acid identity (93.6%) to a gammaproteobacterial *FAD* sequence (SB.02_scaffold_119364_c1_1), which was detected in a microbial metagenome from the sea-ice brine, indicating a likely *FAD*-gene transfer between this gammaproteobacterium-virus pair. Another sea-ice-originated *vFAD* gene (SB_NODE_17_gene-11) clustered with *mFAD* genes from *Bacteroidetes* (*n* = 5) and unclassified bacteria (*n* = 4) in clade B (Fig. 4). These results were further supported by a matching host prediction of the viral population (SB_NODE_17) to *Bacteroidetes* members (Table S7 [40]). Taken together, this evidence supports that this *FAD* was transferred from *Bacteroidetes* to its phage, and that this transfer event was probably host range limited.

The *vFAD* from cryopeg brine formed two distinct clades, A and C, that did not contain any *mFAD* (Fig. 4). Therefore, the origin of the cryopeg *vFAD* is less clear. It is possible that these *FAD* genes have been acquired from low-abundance hosts whose *FAD* genes were not detected in our data set due to the limitation of sequencing depth, or that they have diverged significantly from any potential "source" microbial taxon in

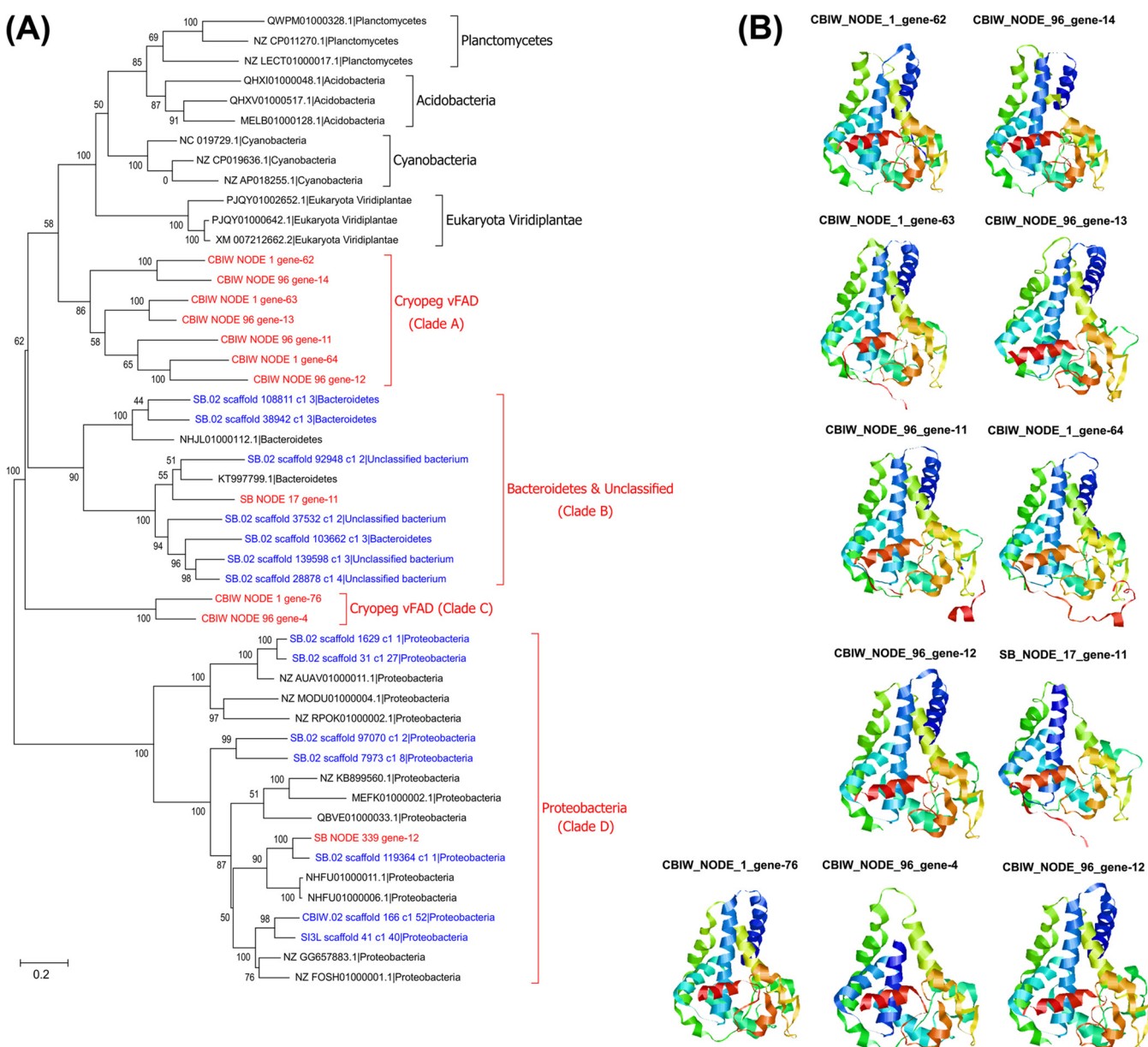

**FIG 4** Diversity and predicted three-dimensional (3D) structures of 11 *vFAD* genes. (A) A neighbor-joining tree was constructed using the predicted nucleotide sequences of 11 *vFAD* genes and some closely associated microbial *FAD* genes. Bootstrap values (expressed as percentages of 1,000 replications) are shown at the branch points. The scale bar indicates a distance of 0.2. *vFAD* sequences are highlighted in red. Microbial *FAD* sequences from Utqiaġvik microbial metagenomes and NCBI nr database are colored in blue and black, respectively. (B) Predicted 3D structures of each vFAD based on the template structure of 4ZYO_A (78).

the ancient cryopeg. Interestingly, all identified cryopeg *vFAD* could be linked to only two viruses (five copies each; Fig. S6 [40]). It is likely that multiple copies of *FAD* in each virus were acquired either from different hosts or from the same host over a long period of time. These results suggest that gene transfers occurred both from host to virus and virus to host multiple times. Our findings are consistent with previous reports of virus-mediated *FAD* gene transfers among microbial hosts (72) as well as numerous photosynthesis genes being transferred within and between cyanobacteria (69, 70, 73).

We next analyzed the evolutionary dynamics of *vFAD* and *mFAD* homologs across lineages (see Materials and Methods). Among the major drivers of microbial evolution is selection pressure. We detected significant differences in the ratio of nonsynonymous to synonymous evolutionary changes ($dN/dS$) ($\omega$) between *vFAD* and *mFAD* clades (Table S10 [40]), indicating that *vFAD* and *mFAD* were under different selection pres-

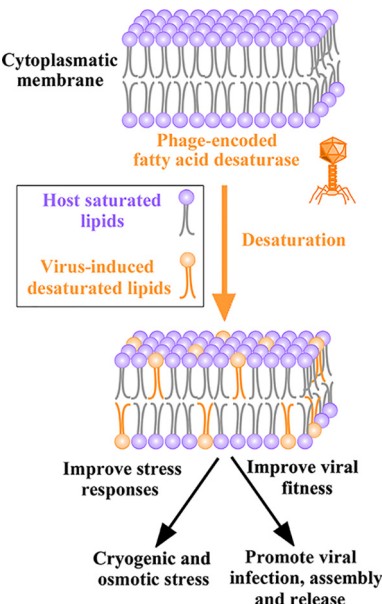

**FIG 5** Potential model for the activity of virus-encoded fatty acid desaturase (vFAD). *vFAD* genes are hypothesized to be expressed upon viral entry into the host. This expression would result in increased unsaturation of fatty acids in the host's cytoplasmic membrane, which would increase membrane fluidity and potentially reduce cryogenic and osmotic stress. Meanwhile, altered membrane fluidity would assist the virus with infection, genome replication, and/or virion release by lysis or membrane budding.

sures. Purifying selection removes potentially deleterious mutations from the genetic background of random neutral mutations. The *vFAD* genes appear to be under purifying selection, implying that they remain functional. The power of this inference depends upon the availability of closely related, known functional cellular homologs: if they are variably available across the data set, such homologs can lead to artifacts (74). However, our evaluation of the data for a large *dS* confirmed that variability was not responsible for the observed differences in selection pressure (Table S11 [40]). Thus, we envisage that the microbes have elevated dosages of *FAD* gene expression that provides advantages to the microbial host, leading to a fitter lineage in nature. This interpretation is in line with previous reports that demonstrated the role of selection pressure in shaping the survival of microbes (75–77).

Indeed, several lines of *in silico* analyses suggested these 12 *vFAD* genes are functional (Fig. 4). First, the deduced amino acid sequences of the 12 *vFAD* revealed that they encoded the well-conserved three histidine cluster motifs (Fig. S8 [40]). Second, 11 of the 12 *vFAD* encoded the three conserved histidine motifs (Fig. S8A [40]), with the 12th *vFAD* (SB_NODE_339_gene-12) encoding other histidine cluster motifs (Fig. S8B [40]). Third, analysis of predicted protein sequences from the 12 *vFAD* using Phyre2 (78) showed that all had 100% confidence scores (except one with 99.8%) that linked them to a stearoyl-coenzyme A (CoA) desaturase, 4ZYO_A, which introduces a double bond at the Δ9 position (the bond between carbons 9 and 10) of saturated fatty stearoyl- or palmitoyl-CoA (79). Based on motif sharing with cyanobacterial FAD proteins and 4ZYO_A, we deduced the *vFAD* from SB_NODE_339_gene-12 might introduce the double bond at carbon Δ6, while the other 11 vFAD might work with carbon Δ9 for desaturation (80–82).

Together, these results suggest that these viruses from cryopeg brine and sea-ice environments obtained *FAD* from their host through HGT and encode AMGs capable of altering host membrane fluidity. Although future experiments are necessary to validate the activity and function of these virus-encoded proteins, and information (e.g., lytic versus lysogenic lifestyles) for these vFAD-carrying phages is limited, the evidence that currently exists suggests they are functional, leading us to hypothesize a conceptual model for their impacts (Fig. 5). Specifically, we propose that *vFAD*-carrying phages help

their hosts to desaturate membrane fatty acids and increase membrane fluidity, and that these changes could increase host tolerance to cold and salt conditions over the course of an infection cycle. For the virus, we posit that this ability alters host membrane structure to promote viral infection and/or virion assembly and release via host lysis or membrane budding, as seen in other systems (83).

**Conclusions.** Polar sea ice and cryopegs within permafrost harbor diverse cold- and salt-tolerant microbes that are considerably different from those in less extreme environments. Unfortunately, global warming is threatening the existence of these extreme environments at an unprecedented pace. We sought here to unlock the secret lives of viruses and microbes in these current and ancient brines threatened by climate change to provide a baseline understanding to assess change (84). This effort revealed abundant viruses that infect dominant microbes that drive the current biogeochemistry of these settings and elucidated the presence and evolutionary history of potentially key genes that impact cold adaptation. Such data-driven inferences establish baseline hypotheses that may be challenging to test due to the extreme nature of these systems but could help to establish ecological impacts of such viruses through emerging technological advances to (i) assess *in situ* activity of microbes and viruses via meta-transcriptomics (85–87) and/or BONCAT-FACS (88), (ii) improve model system experiments to evaluate the altered ecosystem outputs of virus-infected cells (89–91), and (iii) capture microdiverse viral populations and their niche-defining hypervariable genomic islands (e.g., VirION [92]), including capturing RNA viruses (93, 94), where eukaryotes dominate, for example, in alga-rich bottom sea ice. Finally, time series data and greater replication are needed not only for building a model of how life thrives under such extreme conditions but also for evaluating how the underlying mechanisms of gene exchange may change as an environment becomes less extreme.

## MATERIALS AND METHODS

**Site characterization and field sampling. (i) Sea ice.** Snow cover (20 to 40 cm) was removed from the sampling site prior to drilling cores from landfast first-year sea ice in May 2017 near Utqiaġvik, Alaska (71.37311°N, 156.5049°W) (Fig. 1A), using a MARK II ice auger (Kovacs Enterprise). Sea-ice brine was collected by drilling a sackhole (partial core hole) to 75 cm in depth (Fig. 1B). After brine drained into the sackhole for 2 to 3 h, 9 liters of brine was collected into an acid-rinsed (1N HCl) 10-liter Cubitainer by manual pump. A 9- by 117-cm ice core (Fig. 1B) was drilled and sectioned in the field using a 70% ethanol-sterilized custom alloy bow saw to generate three samples: upper sea ice (0 to 25 cm), middle sea ice (50 to 75 cm), and lower sea ice (95 to 117 cm) (Fig. 1B). These ice sections were placed into separate sterile Whirl-Pak bags and transported in an insulated cooler to a cold room (4°C) at the Barrow Arctic Research Center (BARC), where they were processed within 24 h. Each ice section was melted (1:1 volume) into a filtered (0.22-$\mu$m-pore-sized filter) artificial sea salt solution (salinity, 32 g/liter; no. S9883; Sigma).

**(ii) Cryopeg.** Cryopeg brine was encountered by drilling deeper into an existing (previously dry) borehole (9) using a cleaned and ethanol-rinsed ice auger to ~1.5 m below the floor of the Barrow Permafrost Tunnel (71.2944°N, 156.7153°W) (Fig. 1C). A 500-ml sample of cryopeg brine was recovered into a sterile polypropylene bottle, following the use of a specialized apparatus consisting of hand pump, sterile vacuum flask, and sterile tubing extended to the base of the borehole, as previously described (9). The sample was transported in an insulated cooler to a −6°C cold room at BARC and processed in a 4°C cold room within 6 h.

**Viral counting, DNA isolation, and metagenomic sequencing.** Each sample was passed through a polycarbonate 1.2-$\mu$m-pore-size filter (Whatman GF/A) and then a polycarbonate 0.2-$\mu$m-pore-size filter (no. GTTP02500; Isopore) to remove cells and particles of >0.2 $\mu$m. Cells captured on the filters were subjected to DNA extraction and microbial metagenomic sequencing. Virus-like particles (VLP) in the filtrate were counted using the wet-mount method (39). For the sea-ice brine sample, viruses in the filtrate were concentrated using an iron chloride flocculation method (95) and stored at 4°C at BARC. To check for and remove potential contaminants introduced to samples during processing, 100 ml of artificial sea salt solution (named the sea salt control) was used as a background control and was processed in parallel with the field samples through the entire analysis, according to the clean procedures described in a glacier ice core study (96). Concentrated viruses were shipped at 4°C from Utqiaġvik to the Ohio State University in Columbus, where samples were stored at 4°C until DNA isolation.

Approximately half of the volume of each sample was used to isolate genomic DNA as previously described (34). Briefly, the sea-ice brine concentrate was resuspended with ascorbic-EDTA buffer (0.1 M EDTA, 0.2 M MgCl$_2$, 0.2 M ascorbic acid, pH 6.0). All samples were treated with DNase I (100 U/ml) to eliminate free DNA and addition of 100 mM EDTA–100 mM EGTA to halt DNase activity, followed by concentration with a 100-kDa Amicon centrifugal device (no. UFC910096; Millipore), where each sample was resuspended 3 times in an ~800-$\mu$l total volume of ascorbate-EDTA buffer. DNA was then extracted

using Wizard PCR Prep DNA purification resin and minicolumns (no. A7181 and A7211, respectively; Promega, USA) (34). The DNA libraries were prepared using the Accel-NGS 1S Plus DNA library kit (no. 10024; Swift Biosciences) according to the manufacturer's instructions. The samples were sequenced by an Illumina HiSeq 2000 platform ($1\times$ 100 bp) at the JP Sulzberger Genome Center at Columbia University and the sea salt control sample was sequenced at the Joint Genome Institute by Illumina HiSeq platform ($2\times$ 150 bp).

**Physiochemical conditions.** Environmental, geochemical, and nutrient measurements included temperature, salinity, dissolved organic carbon (DOC), $PO_4^{3-}$, $SiO_4^{2-}$, $NO_3^-$, $NO_2^-$, $NH_4^+$, CHN (carbon, hydrogen, and nitrogen), chlorophyll, phaeopigment, particulate exopolysaccharide (pEPS), and dissolved EPS (dEPS) (see Table S1 in the supplemental material in reference 40). The procedures and methods used to obtain these measurements were described in another study (9).

**Microbial community profiles.** Cells captured on the aforementioned filters (0.2-$\mu$m pore size) were used to isolate microbial DNA using a standard phenol-chloroform-isoamyl alcohol extraction method as described previously (97). Extracted DNA was concentrated and purified using the Zymo DNA Clean and Concentrator kit (DCC-5; Zymo Research, Irvine, CA, USA). Library preparation and amplicon sequencing were conducted at MR DNA (Shallowater, TX, USA). Briefly, bar-coded primers 515f/806r (98) were used to amplify the V4 hypervariable region of the 16S rRNA genes of bacteria and archaea for each sample. Amplicons were sequenced using the Illumina MiSeq platform ($2\times$ 300-bp paired-end reads), as described previously (98). Amplicon reads were processed and clustered into operational taxonomic units (OTUs) at 97% sequence similarity (OTU table; Table S5 [40]) using mothur, version 1.40.5 (99), by following the mothur MiSeq SOP (https://mothur.org/wiki/MiSeq_SOP) (100). Taxonomy was assigned to each OTU using the SILVA v132 reference files provided in the mothur wiki. More details for generating the OTU table using mothur were described in another study (9). Finally, the microbial community structure was plotted using the default parameters of the Quantitative Insights Into Microbial Ecology (QIIME, version 1.9.1) software package (101) based on the OTU table.

**Virome analysis.** All metagenomic analyses were supported by the Ohio Supercomputer Center (102). Virome sequence data were processed using the iVirus pipeline with default parameters as described previously (42, 103). Briefly, raw reads of the six viromes, derived from the five field samples and a control sample (sea salt control), were filtered for quality using Trimmomatic v0.36 (104), followed by assembly using metaSPAdes v3.10.1 (k-mer values include 21, 33, and 55) (105) and the prediction of viral contigs using VirSorter v1.0.3 in virome decontamination mode on Cyverse (41). The viral contigs (categories 1 and 2) were first checked for contaminants by comparing them to some viral genomes considered putative laboratory contaminants (e.g., phages cultivated in our laboratory, including *Synechococcus* phages, *Cellulophaga* phages, and *Pseudoalteromonas* phages) using BLASTN and then were clustered into populations if viral contigs shared >95% nucleotide identity across >80% of their lengths, as described previously (42). The longest contig within each population was selected as the seed sequence to represent that population. A coverage table of all viral populations was generated using BowtieBatch and Read2RefMapper by mapping quality-controlled reads to viral populations, and the resulting coverages were normalized by library size to per gigabase of virome (103). Rarefaction curves of cryopeg brine and sea-ice viromes were illustrated by the changing of viral populations along sequencing depth (i.e., read number), which was obtained by subsampling quality-controlled reads (Fig. S1 [40]).

Five populations were obtained from the sea salt control by the same methods. Mapping the clean reads of this control to the 1,306 populations assembled from the five field samples identified only one population in sea salt control (coverage of 20 per gigabase of virome), suggesting that the procedures for handling the field samples were relatively clean. We then mapped the quality-controlled reads of the five field samples to the five sea salt control-assembled populations and detected coverages (range, 0.2 to 106 per gigabase read) for three populations in the field samples, indicating the presence of some contaminants. Contaminants have been detected in many previous investigations of microbial communities, particularly for the low-biomass samples, such as glacier ice (96) and diluted mock microbial communities (106). These contaminants might come from laboratory air, investigators (e.g., human skin and respiratory), and/or tools and reagents used for DNA extraction, PCR amplification, and sequencing (96, 107). Thus, six populations (including five assembled from the sea salt control and one assembled from the field samples) detected in the sea salt control were interpreted as potential contaminants and were removed.

**Analysis and characterization of viral communities.** Principal coordinate analysis (PCoA) was performed using Euclidean distance matrices based on the coverage of each viral population. Associations between the viral and microbial community compositions were evaluated using two-tailed Mantel test by comparing their Euclidean distance matrices.

Taxonomy assignments were performed using vConTACT version 2.0 (46, 47), which groups viral genomes/contigs into viral clusters equivalent to the ICTV genera. Briefly, 15,148 proteins of 476 contigs (length, $\geq$10 kb) and 231,165 proteins from 2,304 archaeal and bacterial viruses (NCBI viral RefSeq, v85) were subjected to all-versus-all BLASTP with an E value threshold of $10^{-4}$ and bit score of 50. Homologous protein clusters then were defined using the Markov clustering algorithm using 2 as an inflation value, as described previously (46). Subsequently, vConTACT v2.0 grouped pairs of closely related genomes and/or contigs into viral clusters based on the number of protein clusters shared between each pair of genomes/contigs using the default parameters (46). The quality-controlled reads of each sample were mapped to the 476 population contigs to obtain the abundance of these contigs in each sample. The positive value of abundance represents the detection of viral population in a sample.

The putative virus-host linkages were predicted *in silico* using nucleotide sequence similarity and sequence composition methods, as described previously (23, 26, 108). The nucleotide sequence of each viral population was compared (BLASTN) to bacterial and archaeal genomes from the NCBI RefSeq

database (release v81). The viral sequences were considered to have a host predicted if they had a bit score of $\geq 50$, E value of $\leq 10^{-3}$, and average nucleotide identity of $\geq 70\%$ across $\geq 2,000$ bp with the host genomes (23). In addition to sequence similarity analyses, we also used the oligonucleotide frequency dissimilarity (by the software VirHostMatcher) measure for host prediction, with ~32,000 bacterial and archaeal genomes as the host database and a dissimilarity score of $\leq 0.3$ as the threshold to pick the host (61).

Auxiliary metabolic genes (AMGs) are virus-encoded "host" genes that may be expressed to reprogram host metabolism during infection and facilitate viral production (21, 24, 109). All 1,305 viral populations ($\geq 5$ kb in size) were examined for virus-encoded AMGs, as described previously (108). Briefly, viral genes from all populations were annotated using a published pipeline (110) by which the open reading frames were predicted for viral contigs using MetaProdigal (111), and sequences were compared to KEGG (112) and UniRef using BLAST and InterProScan (113) using USEARCH (114). The annotations were selected with best-hit matches and a bit score of $\geq 60$. AMGs were first screened by comparison to published AMGs from marine viruses (23) and then screened by manually checking all protein annotations, as described previously (23). Next, sequences of the viral AMG of interest (i.e., *fatty acid desaturase* [*FAD*] gene) were compared to the NCBI nr database (blastp, bit score of $\geq 50$ and E value of $\leq 10^{-3}$) to recruit reference sequences (top 40 hits for each viral AMG sequence, December 2018). In addition, the microbially originated *FAD* genes were extracted from their concurrently sampled microbial metagenomes and combined with previous sequences to study possible *FAD* gene transfers between viruses and their microbial hosts. These analyses obtained 217 and 263 *FAD* genes from metagenomes and NCBI's nr database, respectively. A multiple-sequence alignment of all amino acid sequences was conducted using MAFFT v7.017 (115) with the E-INS-I algorithm and a gap penalty score of 3 for 1,000 iterations. Aligned sequences were subsequently trimmed using TrimAL (116) and manually inspected for overhangs and then used to generate an all-sequence tree. A subset phylogenetic tree then was constructed using representative gene sequences from the all-sequence phylogenetic tree to evaluate the relationship between *vFAD* and microbial *FAD* (*mFAD*). The subset included sequences closely related to the *vFAD*, as well as three randomly selected bacterial and eukaryotic *FAD* representatives from each phylum. Conserved motifs were identified in AMGs based on reported motifs from previous studies (80, 81). Protein sequences from interesting AMGs were structurally modeled using Phyre2 (78) in normal modeling mode to confirm and further resolve functional predictions.

**Sequence analysis and error correction of FAD genes.** Recombination tests were conducted using six programs, RDP (117), MaxChi (118), Chimera (119), SiScan (120), LARD (121), and 3Seq (122), within RDP4 (123). A Bonferroni correction with a *P* value cutoff of 0.05 was applied in each of the tests. A sequence was considered a true recombinant if supported by at least four of the six programs.

**Estimation of stationary nucleotide frequencies along tree branches.** A neighbor-joining (124) tree was reconstructed using the MEGA program, version 6 (125). Evolutionary distances were calculated using the p distances model. Bootstrap resampling using 1,000 replicates was applied to check for internal branch support. The appropriate nucleotide frequency for each branch on the NJ tree was determined by testing two hypotheses: the null hypothesis ($H_0$) and alternative hypothesis ($H_1$). The $H_0$ assigned one set of stationary nucleotide frequencies across all branches, while the $H_1$ assigned different nucleotide frequencies across all branches. The JC69 model (126) was used as the evolutionary model for the null hypothesis. Four independent models (i.e., HKY85, GTR, T92, and F81) were tested for $H_1$. The null and alternative hypotheses assumed that the substitution model followed the HKY85 model across sites with a gamma distribution. Transversion and transition ratios were assumed to be homogeneous among branches. These analyses were conducted using the baseml package in PAML (127). Results are summarized in Table S12 (40).

**Evolutionary rates among lineages.** Evolutionary rates of lineages within the phylogenetic tree were determined based on the local clock model (clock = 0) and the global clock model (clock = 1). The clock models were estimated using the maximum likelihood approach under the nucleotide substitution model HKY85. The evolutionary rates were determined using the baseml package in PAML (127). Statistical comparison of the most suitable clock model was evaluated using the log-likelihood ratio test with a statistical significance level of $\alpha = 0.05$.

**Branch-site selection pressure (dN/dS) analysis.** Selection pressure $\omega$ (*dN/dS*) analysis was carried out on sequences ($n = 55$) from the cryopeg brine (i.e., CBIW), where the majority of *vFAD* genes (10 of 12) were obtained. Selection pressure was measured using codon models with maximum likelihood estimated with the codeml package in PAML (127). Three hypotheses of possible selection pressure scenarios were compared, including $H_0$ (model = 0, NsSite = 0) and equal selection pressure between *mFAD* and *vFAD* clades. The alternative hypothesis was independent selection pressure on the *vFAD* and *mFAD* clades (model = 2, NsSite = 2). Statistical comparison of the most suitable hypothesis was evaluated using the log-likelihood ratio test with a statistical significance level of $\alpha = 0.05$.

**Data availability.** Virome reads and putative viruses were deposited in Cyverse (https://doi.org/10.25739/0e7n-6611). The amplicon data of 16S rRNA genes are publicly available in the Sequence Read Archive individually with accession numbers SRR9070059–SRR9070073 under BioProject accession number PRJNA540708.

## ACKNOWLEDGMENTS

This work was funded by the Gordon and Betty Moore Foundation grant number 5488 and also supported by the Byrd Polar and Climate Research Center Postdoctoral Fellowship. A portion of this research was performed under the JGI-EMSL Collaborative

Science Initiative and used resources at the DOE Joint Genome Institute and the Environmental Molecular Sciences Laboratory, which are DOE Office of Science User Facilities. Both facilities are sponsored by the Office of Biological and Environmental Research and operated under contract no. DE-AC02-05CH11231 (JGI) and DE-AC05-76RL01830 (EMSL).

We greatly appreciate the help provided by Maria C. Gazitúa, Simon Roux, Ann C. Gregory, Gareth Trubl, and Dean R. Vik with virome analysis, by Sergei A. Solonenko with evolutionary analysis, and by Max Showalter, Hannah Dawson, Go Iwahana, Jodi Young, and Anders Torstensson with sample collection. We are grateful to the Ukpeagvik Iñupiat Corporation Science team for expert logistics support and access to the permafrost tunnel. Custom saw blades were kindly provided by Paul Shemeta of Diggit Garden Tools. Bioinformatics were supported by the Ohio Supercomputer Center.

We have no competing interest to declare.

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
