## [Reviewer comments · mSystems]

Viral ecogenomics of Arctic cryopeg brine and sea ice

Zhi-Ping Zhong, Josephine Rapp, James Wainaina, Natalie Solonenko, Heather Maughan, Shelly Carpenter, Zachary Cooper, Ho Bin Jang, Benjamin Bolduc, Jody Deming, and Matthew Sullivan
Corresponding Author(s): Matthew B Sullivan, The Ohio State University

Review Timeline:

Submission Date:	March 19, 2020
Editorial Decision:	April 27, 2020
Revision Received:	May 18, 2020
Accepted:	May 24, 2020

Editor: Joanne Emerson

Reviewer(s): Disclosure of reviewer identity is with reference to reviewer comments included in decision letter(s). The following individuals involved in review of your submission have agreed to reveal their identity: Maureen Berg (Reviewer #2)

Transaction Report:

DOI: <https://doi.org/10.1128/mSystems.00246-20>

April 27, 2020

Dr. Zhi-Ping Zhong
The Ohio State University
Byrd Polar and Climate Research Center
Columbus, Ohio

Re: mSystems00246-20 (Viral ecogenomics of Arctic cryopeg brine and sea ice)

Dear Dr. Zhi-Ping Zhong:

Your manuscript has been reviewed by two experts, and I am pleased to report that both found it to be an important contribution to the field. Pending minor revisions in response to the reviewers' comments below, the manuscript is in principle publishable in mSystems.

Below you will find the comments of the reviewers.

To submit your modified manuscript, log onto the eJP submission site at <https://msystems.msubmit.net/cgi-bin/main.plex>. If you cannot remember your password, click the "Can't remember your password?" link and follow the instructions on the screen. Go to Author Tasks and click the appropriate manuscript title to begin the resubmission process. The information that you entered when you first submitted the paper will be displayed. Please update the information as necessary. Provide (1) point-by-point responses to the issues raised by the reviewers as file type "Response to Reviewers," not in your cover letter, and (2) a PDF file that indicates the changes from the original submission (by highlighting or underlining the changes) as file type "Marked Up Manuscript - For Review Only."

Due to the SARS-CoV-2 pandemic, our typical 60 day deadline for revisions will not be applied. I hope that you will be able to submit a revised manuscript soon, but want to reassure you that the journal will be flexible in terms of timing, particularly if experimental revisions are needed. When you are ready to resubmit, please know that our staff and Editors are working remotely and handling submissions without delay. If you do not wish to modify the manuscript and prefer to submit it to another journal, please notify me of your decision immediately so that the manuscript may be formally withdrawn from consideration by mSystems.

To avoid unnecessary delay in publication should your modified manuscript be accepted, it is important that all elements you upload meet the technical requirements for production. I strongly recommend that you check your digital images using the Rapid Inspector tool at <http://rapidinspector.cadmus.com/RapidInspector/zmw/>.

Corresponding authors may join or renew ASM membership to obtain discounts on publication fees. Need to upgrade your membership level? Please contact Customer Service at

Service@asmusa.org.

Thank you for submitting your work to mSystems.

Sincerely,

Joanne Emerson

Editor, mSystems

Journals Department
Reviewer comments:

Reviewer #1 (Comments for the Author):

SUMMARY AND RELEVANCE

The authors used low-input, quantitative viral metagenomes from icy brine environments, including sea ice brine, cryopeg, and melted sea ice, to characterize viral populations, infected hosts from these environments, and functional genes encoded by viruses such as fatty acid desaturase (FAD) that would be relevant to hosts' survival under cold and salt stress. These novel data from viruses in extreme environments are useful in their own right for better characterizing organisms in understudied extreme environments, especially icy environments endangered by climate change that may provide insight into life in past climates on Earth (e.g. "snowball earth"). Inclusion of the data on host infection and functional genes makes this manuscript more powerful by implying specific host interactions of viruses in these communities, including new information on potential constraints on the apparent range of virus-mediated horizontal gene transfer (HGT) of auxiliary metabolic genes (AMG) in extreme environments.

GENERAL COMMENTS

The manuscript combines a number of recently developed tools in viromics to assemble viral contigs, build phylogenies, and assign potential function to possible host acquired genes, along with their relationships to potential hosts' genomes, to draw the most holistic inference possible from a limited sample set regarding their viral ecology.

A major drawback of the study is the small sample size with no replication of the environment types. However, I understand well how difficult samples from extreme environments can be to obtain, and much of our understanding of them must therefore be extrapolated from regrettably small sample sizes due to logistics. The study nevertheless provides new insights to be expanded on in future research.

The conclusions in by this manuscript are sound and well-supported by appropriate analyses of the data. Two key findings in my view merit some more discussion:

First, the fact that cryopegs are introduced as having been isolated for 10k years makes it seem surprising that any viral clusters (VC) were shared at all between the cryopeg brine and sea ice. Even though viruses within VC may still be rather distantly related, would any overlap be expected at this taxonomic resolution after that amount of time? Especially given that four of the five were actually shared between the cryopeg brine and the sea ice - but not the more concentrated sea-ice brine (did I understand that correctly?). Further discussion of whether this is indeed surprising or rather expected, or the context of whether these viruses are likely to still be infections, or which potential related hosts may be shared among the environments, would seems to be appropriate here.

A second area that merits more discussion is any further detail available on life history of the four viral populations found to encode FAD genes. The left-hand arrow in Fig 5 could represent selection on the vFAD genes not only through short-term increases in production, but in long-term host fitness in a lysogenic infection (or some degree of delayed lysis of the host that permits host reproduction, whether or not the virus integrates into the host's genome). Were any of these four viral populations identified in the group of prophages? Or might they remain outside the host genome without actively replicating for many generations? Or were any of these populations ones that were linked to known viral groups whose life history could be discussed? If none of these links apply to the four viral populations of interest, at least stating that would clarify for the reader that no further clues are available and overlooked.

Specific comments follow where minor clarifications would improve the manuscript.

SPECIFIC COMMENTS

L68: What is a "temperature-dependent volume fraction?" Does this mean that some fraction of these environments' volumes is a liquid brine, and that fraction depends on the temperature? Consider clarifying the wording.

L90-91: Do these same phyla also dominate source water samples, just with different genera or species, though? If so, make this distinction clear. Or is the rest of the paragraph not intended to follow from the contrast set up between source and brine waters in the topic sentence, but only to introduce dominant taxa in these environments?

L100: Have viruses in brines been previously characterized at all to your knowledge?

L126-127: Given the framing of sea-ice brine as distinct from bulk sea ice in the intro, it would be nice to present the VLP count for sea-ice brine as distinct from those of the melted bulk sea ice if those data are available.

L129: Does the phrase "these samples" refer to all five samples, or just the sea ice-derived samples from the previous sentence?

L169-172: The fact that cryopegs are introduced as having been isolated for 10k years makes it seem astonishing that ANY viral clusters (VC) were shared at all between the cryopeg brine and

sea ice! Even though viruses within VC may still be rather distantly related, this overlap is actually surprising, especially that four of the five were actually shared between the cryopeg brine and the sea ice - but not the sea-ice brine? No discussion of whether this is surprising, or the context of whether these viruses are likely to still be active, or which potential hosts may be shared among the environments?

L254: Why the transition word "similarly?" A relative abundance of 1.3% sounds much lower than the previously mentioned relative high abundance of 12.9%, and numbers for other samples are not given to clarify if this is a high relative abundance for Colwellia-linked viruses within the data set. For this entire paragraph, consider including some information on rank abundance rather than just raw relative abundance in the text here so that readers do not have to chase down the supplemental tables and figures to see how these taxa compare to others across samples to evaluate the claims. Or make a plot correlating relative abundances (or rank abundances) of viruses with their potential predicted hosts?

L259-261: I am unclear what the argument being advanced in this sentence is: that the high relative abundance of viruses relative to their hosts is due to high specificity of viruses that attack this lineage? I suggest clarifying the wording.

L329-331: Purifying selection mitigated what? Differences in selection pressures? In what way? An additional sentence clarifying the proposed mechanism would improve this point.

L333-336: Were any of the four viral populations containing FAD genes included in the relatively few viral populations that were identified as prophages? Were any of the four viral populations with FAD genes related to any known viruses whose life history could be discussed? Possibly a missed opportunity here.

L378-381: Perhaps also mention the need for greater replication to characterize the variability within these environments, given the limited number of samples analyzed here from which to draw inference?

L398-402: Although the previous study is cited for methods of sampling the cryopeg, it would be nice to at least mention here how the liquid was accessed from within the tunnel - whether it was accessed via a previously drilled hole in the tunnel floor, or drilled with a sterilized instrument specifically for this sampling.

Reviewer #2 (Comments for the Author):

I thought this paper was extremely well written, and the science is solid. I always have problems with lack of replicates with many metagenome papers, but I felt like the analyses and conclusions drawn from these data are measured, appropriate, and ultimately incredibly interesting. The only comment I really have is that I feel like Figure 3C is unnecessary; Figure 3B is interesting by showing how different the lower sea ice is compared to the middle and upper portions (and compared to the brine samples). However, Figure 3C is not really showing anything Figure 3B isn't, and with no replication, the conclusions you can draw from these "small" differences between samples in ordination plots is limited.

Editor's summary comments:

Your manuscript has been reviewed by two experts, and I am pleased to report that both found it to be an important contribution to the field. Pending minor revisions in response to the reviewers' comments below, the manuscript is in principle publishable in mSystems.

Response: We appreciate for your time and efforts and those of the referees during these challenging times. Please find a point-by-point response to the reviews below, where the reviewer's original comments are in black and our responses are in blue text.

Reviewer #1 (Comments for the Author):

SUMMARY AND RELEVANCE

The authors used low-input, quantitative viral metagenomes from icy brine environments, including sea ice brine, cryopeg, and melted sea ice, to characterize viral populations, infected hosts from these environments, and functional genes encoded by viruses such as fatty acid desaturase (FAD) that would be relevant to hosts' survival under cold and salt stress. These novel data from viruses in extreme environments are useful in their own right for better characterizing organisms in understudied extreme environments, especially icy environments endangered by climate change that may provide insight into life in past climates on Earth (e.g. "snowball earth"). Inclusion of the data on host infection and functional genes makes this manuscript more powerful by implying specific host interactions of viruses in these communities, including new information on potential constraints on the apparent range of virus-mediated horizontal gene transfer (HGT) of auxiliary metabolic genes (AMG) in extreme environments.

GENERAL COMMENTS

The manuscript combines a number of recently developed tools in viromics to assemble viral contigs, build phylogenies, and assign potential function to possible host acquired genes, along with their relationships to potential hosts' genomes, to draw the most holistic inference possible from a limited sample set regarding their viral ecology.

A major drawback of the study is the small sample size with no replication of the environment types. However, I understand well how difficult samples from extreme environments can be to obtain, and much of our understanding of them must therefore be extrapolated from regrettably small sample sizes due to logistics. The study nevertheless provides new insights to be expanded on in future research.

Response #1: We appreciate your awareness and understanding of how difficult it can be to obtain any samples in these challenging extreme environments. Much care and attention went into the field program to obtain even a limited number of samples, and sometimes (as it seems you understand) that is lost on reviewers.

The conclusions in by this manuscript are sound and well-supported by appropriate analyses of the data. Two key findings in my view merit some more discussion:

First, the fact that cryopegs are introduced as having been isolated for 10k years makes it seem surprising that any viral clusters (VC) were shared at all between the cryopeg brine

and sea ice. Even though viruses within VC may still be rather distantly related, would any overlap be expected at this taxonomic resolution after that amount of time?

Especially given that four of the five were actually shared between the cryopeg brine and the sea ice - but not the more concentrated sea-ice brine (did I understand that correctly?). Further discussion of whether this is indeed surprising or rather expected, or the context of whether these viruses are likely to still be infections, or which potential related hosts may be shared among the environments, would seem to be appropriate here.

Response #2: Thank you for highlighting the need for more discussion to clarify this point. Our expectation is that shared viral VCs/genera between cryopeg and sea-ice environments represent genus-level taxonomy that reflects very broad characteristics of viral replication and packaging (1). Even though long separation from sea ice, cryopegs were also originally sourced from Arctic seawater (2) and continue to share some common physical properties (subzero temperature and hyper-salinity), so that a sharing of some broad viral characteristics is not unexpected. For ecological relevance, we asked whether viral populations (i.e., species (3, 4)) were shared across the cryopeg and sea-ice environments, as this level of taxonomy is thought to represent the ecologically relevant (rather than taxonomically relevant) unit (5). We found near-completely different viral communities at this level. Viral types at a higher taxonomic level are shared across these samples, but at the level of ecological units they are highly divergent, as one would expect based upon their microbial and physicochemical profiles. We have added the following short discussion in the new version (Lines 175–178):

“The shared VCs suggest similarities in viral reproduction, as genus-level viral communities typically reflect the broad characteristics of viral replication and packaging (45), as opposed to ecological distinctions (see below).”

We apologize for the confusion leading to the second question. We did not mean “four of the five were actually shared between the cryopeg brine and the sea ice - but not the more concentrated sea-ice brine”. We found that five VCs were shared between cryopeg and sea-ice environments, where the latter refers to both sea ice sections and sea ice brine. Four of these five are previously undescribed genera (i.e., these four VCs contained novel viral populations found only from cryopeg and sea-ice environments), while one was assigned to a genus within the *Podoviridae* (i.e., this VC contained viruses already present in the RefSeq database). We have edited these sentences in the new version (Lines 178–182) to clarify these points, as follows:

“Four of these five shared VCs contained viral populations exclusive to cryopeg brine and sea-ice environments (i.e., not available in the RefSeq database; Fig. S2), while the other one included sequences most similar to a temperate Myxococcus phage Mx8 (VC_248_0, Table S2) in the RefSeq database (47).”

A second area that merits more discussion is any further detail available on life history of the four viral populations found to encode FAD genes. The left-hand arrow in Fig 5 could represent selection on the vFAD genes not only through short-term increases in production, but in long-term host fitness in a lysogenic infection (or some degree of delayed lysis of the host that permits host reproduction, whether or not the virus

integrates into the host's genome). Were any of these four viral populations identified in the group of prophages? Or might they remain outside the host genome without actively replicating for many generations? Or were any of these populations ones that were linked to known viral groups whose life history could be discussed? If none of these links apply to the four viral populations of interest, at least stating that would clarify for the reader that no further clues are available and overlooked.

Response #3: Thank you for these suggestions. We used two tools (VirSorter (6) and PHACTS (7)) and manually checked for gene annotations (e.g., integrase genes) to identify prophages from these four viral populations. However, none of these populations was identified as prophage by these efforts, or by inference from the VC-based taxonomic assignment (i.e., not close enough to a known temperate phage reference genome). We interpret these results to mean that we captured some combination of novel temperate phages that are not identifiable as such, along with viruses derived from actively lytic infections, which limits further discussion for these populations. We have clarified this point in the revised manuscript as follows (Lines 370–374):

“Though future experiments are necessary to validate the activity and function of these virus-encoded proteins, and information (e.g., lytic versus lysogenic life styles) for these vFAD-carrying phages is limited, the evidence that currently exists suggests they are functional, leading us to hypothesize a conceptual model for their impacts (Fig. 5).”

Specific comments follow where minor clarifications would improve the manuscript.

SPECIFIC COMMENTS

L68: What is a "temperature-dependent volume fraction?" Does this mean that some fraction of these environments' volumes is a liquid brine, and that fraction depends on the temperature? Consider clarifying the wording.

Response #4: Yes, that is the correct meaning. These environments contain some liquid volume (brine) even at very low temperatures (e.g., below the freezing point of pure water), due to the high salinity of the brine. The portion of these frozen environments that is brine-filled depends on the temperature: increasing the temperature increases the “volume fraction” (a standard phrase in the geophysical literature for the percentage of a frozen matrix that is liquid) and simultaneously decreases the salinity of the brine (as more frozen water moves into the liquid phase); decreasing the temperature shrinks the volume fraction and simultaneously increases its salinity. These liquids only freeze if the temperature is decreased dramatically, to the eutectic point of the dissolved salts. To avoid confusion around the geophysical phrase we used, we have revised this sentence in the new version (Lines 67–70). It now reads as:

“Each of these frozen environments contains inhabitable brines that remain liquid below the freezing point of water due to their high salt concentrations.”

L90-91: Do these same phyla also dominate source water samples, just with different genera or species, though? If so, make this distinction clear. Or is the rest of the

paragraph not intended to follow from the contrast set up between source and brine waters in the topic sentence, but only to introduce dominant taxa in these environments?

Response #5: We apologize for the confusion in these sentences. Our intent was simply to introduce previously reported microbial communities in these environments. While these phyla are also dominant in arctic seawater (i.e., source water) according the literature, we did not intend to discuss this topic here. We have revised the first sentence of this paragraph (Line 87–88) in the new manuscript as following:

“Microbial communities in sea ice and cryopegs have been investigated previously in many studies.”

L100: Have viruses in brines been previously characterized at all to your knowledge?

Response #6: We had introduced and referenced this background on subzero brines in the manuscript on original Lines 106–109. Briefly, there is one report of viruses in cryopeg brine (viral counts and one virome of very shallow sequencing depth) and numerous (largely microscopy- and culture-based) studies of viruses in sea ice. However, none of these studies used a modern viral ecogenomics toolkit to study the viral communities and their ecological role in these extreme environments. We revised the main text (Lines 107–111) to include further reference to a summary table of this prior work, which now reads as:

“Although viruses are known to be abundant in sea ice (16, 19, 32), cryopeg brines (15), and other subzero brines (as summarized in (15), their Table 2), no study has used a modern viral ecogenomics toolkit (22) to study the viral communities and their ecological role in these extreme environments, due in part to limited sample volumes and availability.”

L126-127: Given the framing of sea-ice brine as distinct from bulk sea ice in the intro, it would be nice to present the VLP count for sea-ice brine as distinct from those of the melted bulk sea ice if those data are available.

Response #7: To count VLP in this study, we used a recently-developed wet-mount method (8) that provides accurate counting if the viral concentration is $\geq 1 \times 10^6 \text{ ml}^{-1}$. Below that threshold, the counting error is high (± 5 times). Unexpectedly, the viral counts of all of the sea ice samples (sea ice brine and melted ice sections) fell below 10^6 ml^{-1} , nor had we opted to use sample volume for a sample concentration step, reserving precious volume for the sequencing effort. Although we did expect higher counts in the brine based on prior work showing bacterial concentration in the brine phase of sea ice (9), we cannot distinguish the sea ice brine versus melted bulk sea ice VLP counts given the associated error. In the text, we therefore only highlighted the great difference (by orders of magnitude) between VLP concentration in cryopeg versus sea-ice environments. We acknowledge that accurate viral counts could be obtained in future using larger volumes of sea-ice samples. We also acknowledge that we failed to cite Table S4 at this point in the text, where the VLP counts are provided, along with a footnote indicating the associated error. In the revised version (Line 130), we have added this citation, which also obliged changing the numbering system of the supplemental tables (Table S4

became Table S1, and Tables S2–S4 shifted accordingly; the numbering of Tables S5 and beyond did not require renumbering).

L129: Does the phrase "these samples" refer to all five samples, or just the sea ice-derived samples from the previous sentence?

Response #8: Thank you for pointing out this ambiguity. The phrase “these samples” was meant to refer only to the sea ice-derived samples, and has been revised on Lines 130–132 of the manuscript as suggested:

“Similarly, the concentrations of viral DNA extracted from the sea ice-derived samples were also below the detection limit (< 0.1 ng/μl) of Qubit 3.0 (Thermo Fisher Scientific).”

L169-172: The fact that cryopegs are introduced as having been isolated for 10k years makes it seem astonishing that ANY viral clusters (VC) were shared at all between the cryopeg brine and sea ice! Even though viruses within VC may still be rather distantly related, this overlap is actually surprising, especially that four of the five were actually shared between the cryopeg brine and the sea ice - but not the sea-ice brine? No discussion of whether this is surprising, or the context of whether these viruses are likely to still be active, or which potential hosts may be shared among the environments?

Response #9: Please see our response (i.e., Response #2) to these questions where they also appear under General Comments.

L254: Why the transition word "similarly?" A relative abundance of 1.3% sounds much lower than the previously mentioned relative high abundance of 12.9%, and numbers for other samples are not given to clarify if this is a high relative abundance for *Colwellia*-linked viruses within the data set. For this entire paragraph, consider including some information on rank abundance rather than just raw relative abundance in the text here so that readers do not have to chase down the supplemental tables and figures to see how these taxa compare to others across samples to evaluate the claims. Or make a plot correlating relative abundances (or rank abundances) of viruses with their potential predicted hosts?

Response #10: We apologize for the difficulty in following our results in these sentences. The topic of this paragraph is to argue that some dominant viruses infected the dominant microbial groups (at genus level). Thus we used relative abundance to present the dominant genera here (i.e., *Marinobacter*, *Glaciecola*, and *Colwellia*). The word “similarly” was used as both *Marinobacter* and *Glaciecola* had higher virus relative abundance than their (*in silico* predicted) host relative abundance, while *Colwellia* had higher host relative abundance. In rewording the text (and reordering the results), we removed “similarly”. We have also provided the rank abundance information for the viral populations with host predictions on Lines 253–271 in the revised manuscript. Now these sentences read as:

*“The predicted host genera (identified by the sequence similarity method) that were most abundant in these samples included *Marinobacter*, *Glaciecola*, and*

Colwellia (Table S6), many members of which are tolerant of low temperature and high salinity as discussed above. The relative abundance of *Marinobacter*-associated viral populations was high (12.9%) in cryopeg brine (Table S6), including one (i.e., CBIW_NODE_98) with the third-highest coverage (coverage = 684) among all populations in this sample (Table S4). This finding is consistent with the dominance (53.0%) of the bacterial genus *Marinobacter* in this cryopeg brine sample (Fig. S4), and in other samples of cryopeg brine from the same location (10). The *Glaciecola*-associated virus SI3M_NODE_10 was the second-most abundant (relative abundance 14.8–18.0%) viral population in both middle and upper sea ice viromes (Table S4 & S6), while members of *Glaciecola* only accounted for 1.4–1.9% of microbial profiles in these samples (Fig. S4). *Colwellia*-linked virus SI3M_NODE_1 was the sixth-most abundant viral population and had a relative abundance of 1.3% in middle sea ice, where the microbial community was dominated by the genus *Colwellia* with 8.3% of relative abundance (Fig. S4).”

L259-261: I am unclear what the argument being advanced in this sentence is: that the high relative abundance of viruses relative to their hosts is due to high specificity of viruses that attack this lineage? I suggest clarifying the wording.

Response #11: Thank you for bringing this problem to our attention, as further study indicated that our argument was not supported; i.e., the high relative-abundance virus/host ratio could not be argued to reflect viral activity. Thus, we removed this sentence from the revised manuscript (Lines 271–273).

L329-331: Purifying selection mitigated what? Differences in selection pressures? In what way? An additional sentence clarifying the proposed mechanism would improve this point.

Response #12: We apologize for these confused sentences. The following sentences have been added in the manuscript for further clarification (Lines 345–354).

“Purifying selection removes potentially deleterious mutations from among the genetic background of random neutral mutations. The vFAD genes appear to be under purifying selection, implying that they likely remain functional. The power of this inference depends upon the availability of closely related, known functional cellular homologs: if variably available across the dataset, such homologs can lead to artifacts. However, our evaluation of the data for a large dS, confirmed that variability was not responsible for the observed differences in selection pressure (Table S11). We thus envisage that the microbes have elevated dosage of FAD gene expression that provides advantages to the microbial host, leading to a fitter lineage in nature”

L333-336: Were any of the four viral populations containing FAD genes included in the relatively few viral populations that were identified as prophages? Were any of the four

viral populations with FAD genes related to any known viruses whose life history could be discussed? Possibly a missed opportunity here.

Response #13: We appreciate that you brought this possibility to our attention. Please see our response (i.e., Response #3) to these questions where they also appear under General Comments.

L378-381: Perhaps also mention the need for greater replication to characterize the variability within these environments, given the limited number of samples analyzed here from which to draw inference?

Response #14: This need was added to the revised manuscript (Line 398). Now it reads as:

“Finally, time series data and greater replication are needed not only for building a model of how life thrives under such extreme conditions, but also for evaluating how the underlying mechanisms of gene exchange may change as an environment becomes less extreme.”

L398-402: Although the previous study is cited for methods of sampling the cryopeg, it would be nice to at least mention here how the liquid was accessed from within the tunnel - whether it was accessed via a previously drilled hole in the tunnel floor, or drilled with a sterilized instrument specifically for this sampling.

Response #15: We have added this information to the revised version on Lines 418-426, which now read as:

“Cryopeg brine was encountered by drilling deeper into an existing (previously dry) borehole (9) using a cleaned and ethanol-rinsed ice auger to ~ 1.5 m below the floor of the Barrow Permafrost Tunnel (71.2944 °N, 156.7153 °W; Fig. 1C). A 500-ml sample of cryopeg brine was recovered into a sterile polypropylene bottle following use of a specialized apparatus consisting of hand pump, sterile vacuum flask and sterile tubing extended to the base of the borehole as previously described (9). The sample was transported in an insulated cooler to a –6 °C cold room at BARC, and processed in a 4°C cold room within 6 hours.”

Reviewer #2 (Comments for the Author):

I thought this paper was extremely well written, and the science is solid. I always have problems with lack of replicates with many metagenome papers, but I felt like the analyses and conclusions drawn from these data are measured, appropriate, and ultimately incredibly interesting. The only comment I really have is that I feel like Figure 3C is unnecessary; Figure 3B is interesting by showing how different the lower sea ice is compared to the middle and upper portions (and compared to the brine samples). However, Figure 3C is not really showing anything Figure 3B isn't, and with no replication, the conclusions you can draw from these "small" differences between samples in ordination plots is limited.

Response: We appreciate your kind words about our work, and thank you for the comments. We understand and agree with the comments about Figure 3C. We have removed it from the new manuscript, and revised the text and figure legend accordingly.

References:

1. Jang HB, Bolduc B, Zablocki O, Kuhn JH, Roux S, Adriaenssens EM, Brister JR, Kropinski AM, Krupovic M, Lavigne R, Turner D, Sullivan MB. 2019. Taxonomic assignment of uncultivated prokaryotic virus genomes is enabled by gene-sharing networks. *Nature Biotechnology* 37:632-639.
2. Colangelo-Lillis J, Eicken H, Carpenter SD, Deming JW. 2016. Evidence for marine origin and microbial-viral habitability of sub-zero hypersaline aqueous inclusions within permafrost near Barrow, Alaska. *FEMS Microbiol Ecol* 92:fiw053.
3. Gregory AC, Zayed AA, Conceicao-Neto N, Temperton B, Bolduc B, Alberti A, Ardyna M, Arkhipova K, Carmichael M, Cruaud C, Dimier C, Dominguez-Huerta G, Ferland J, Kandels S, Liu Y, Marec C, Pesant S, Picheral M, Pisarev S, Poulain J, Tremblay JE, Vik D, Tara Oceans C, Babin M, Bowler C, Culley AI, de Vargas C, Dutilh BE, Iudicone D, Karp-Boss L, Roux S, Sunagawa S, Wincker P, Sullivan MB. 2019. Marine DNA viral macro- and microdiversity from pole to pole. *Cell* 177:1109-1123.
4. Gregory AC, Solonenko SA, Ignacio-Espinoza JC, LaButti K, Copeland A, Sudek S, Maitland A, Chittick L, dos Santos F, Weitz JS, Worden AZ, Woyke T, Sullivan MB. 2016. Genomic differentiation among wild cyanophages despite widespread horizontal gene transfer. *Bmc Genomics* 17.
5. Duhaime MB, Solonenko N, Roux S, Verberkmoes NC, Wichels A, Sullivan MB. 2017. Comparative omics and trait analyses of marine *Pseudoalteromonas* phages advance the phage OTU concept. *Frontiers in Microbiology* 8.
6. Roux S, Enault F, Hurwitz BL, Sullivan MB. 2015. VirSorter: mining viral signal from microbial genomic data. *PeerJ* 3:e985.
7. McNair K, Bailey BA, Edwards RA. 2012. PHACTS, a computational approach to classifying the lifestyle of phages. *Bioinformatics* 28:614-8.
8. Cunningham BR, Brum JR, Schwenck SM, Sullivan MB, John SG. 2015. An inexpensive, accurate, and precise wet-mount method for enumerating aquatic viruses. *Appl Environ Microbiol* 81:2995-3000.

9. Deming JW. 2010. Sea ice bacteria and viruses, p 247-282. *In* Thomas DN, Dieckmann GS (ed), *Sea Ice*, 2nd ed
doi:<https://doi.org/10.1002/9781444317145.ch7> Wiley-Blackwell, Oxford, UK.

May 24, 2020

Dr. Zhi-Ping Zhong
The Ohio State University
Byrd Polar and Climate Research Center
Columbus, Ohio

Re: mSystems00246-20R1 (Viral ecogenomics of Arctic cryopeg brine and sea ice)

Dear Dr. Zhi-Ping Zhong:

I am pleased to report that your manuscript has been accepted for publication in mSystems, and I am forwarding it to the ASM Journals Department for publication. For your reference, ASM Journals' address is given below. Before it can be scheduled for publication, your manuscript will be checked by the mSystems senior production editor, Ellie Ghatineh, to make sure that all elements meet the technical requirements for publication. She will contact you if anything needs to be revised before copyediting and production can begin. Otherwise, you will be notified when your proofs are ready to be viewed.

Sincerely,

Joanne Emerson
Editor, mSystems

Journals Department
Phone: 1-202-942-9338